# Plasma Exchange May Enhance Antitumor Effects by Removal of Soluble Programmed Death-Ligand 1 and Extracellular Vesicles: Preliminary Study

**DOI:** 10.3390/biomedicines10102483

**Published:** 2022-10-05

**Authors:** Kazumasa Oya, Larina Tzu-Wei Shen, Kazushi Maruo, Satoshi Matsusaka

**Affiliations:** 1Department of Dermatology, Faculty of Medicine, University of Tsukuba, Tsukuba 305-8575, Japan; 2Department of Clinical Research and Regional Innovation, Faculty of Medicine, University of Tsukuba, Tsukuba 305-8587, Japan; 3Department of Biostatistics, Faculty of Medicine, University of Tsukuba, Tsukuba 305-8587, Japan; 4Tsukuba Clinical Research and Development Organization, University of Tsukuba, Tsukuba 305-8587, Japan

**Keywords:** antibody-drug conjugate, extracellular vesicles, plasma exchange, trastuzumab emtansine

## Abstract

The antitumor effect of antibody-drug conjugates (ADC) is the main factor in achieving cures. Although the mechanism of tumor resistance to treatment is multifaceted, tumor-derived extracellular vesicles (T-EVs) have been implicated as contributing to the attenuation of ADC therapeutic efficacy. Thus, strategies to eliminate T-EVs are highly promising for overcoming drug resistance. Here we demonstrate plasma exchange therapy to remove T-EVs, decreasing their amount in vitro by 75%. Although trastuzumab emtansine (T-DM1) treatment alone was effective in our rat tumor model, the combination therapy of T-DM1 and T-EV filtration achieved early tumor shrinkage. Our results indicate that T-EV filtration plus ADC is a promising strategy for overcoming drug resistance.

## 1. Introduction

Trastuzumab emtansine (T-DM1) is an antibody-drug conjugate (ADC) designed with an anti-human epidermal growth factor receptor 2 (HER2) antibody conjugated with an inhibitor of microtubule polymerization, DM1 [1]. T-DM1 inhibits tumor growth by inhibiting growth signaling through the HER2 pathway. In addition, the binding of T-DM1 to HER2 (highly expressed in cancer cells) leads to the intracellular incorporation of T-DM1 and apoptosis via the activation and release of DM1 [1]. Although clinical trials demonstrated that T-DM1 improves survival in patients with HER2-positive metastatic breast cancer, a significant fraction of patients did not respond to T-DM1 [2,3]. Thus, a novel approach to enhance the antitumor effect of T-DM1 is necessary.

Programmed death (PD-1) is a checkpoint inhibitory molecule mainly expressed in, activated in, and serving as a marker of exhausted T cells [4]. The binding of its ligand, programmed death-ligand 1 (PD-L1), to PD-1 in T cells transduces inhibitory signals, resulting in the attenuation of T cell proliferation, cytokine production, and cytotoxic molecule production [4]. Therefore, the PD-1/PD-L1 axis is a primary mechanism of escape from antitumor immune responses [4]. PD-L1 not only localizes on the tumor cell surface but also exists in the blood as soluble variants which are generated by the shedding of membrane-type PD-L1 via metalloproteinase activity [5]. A phase two, multicenter, randomized study revealed that median progression-free survival was almost half in the PD-L1-positive breast cancer subgroup compared with the negative group after T-DM1 treatment [6]. On the other hand, the concentration of soluble PD-L1 (sPD-L1) is associated with the tumor stage in breast cancer [7]. Furthermore, higher sPD-L1 is reported to be associated with shorter overall survival in breast cancer patients after trastuzumab treatment [8], indicating that PD-L1 may ablate the antitumor effect of trastuzumab and T-DM1 in breast cancer. PD-L1 also exists on extracellular vesicles (EVs), which are characterized by lipid bilayer small particles from diverse types of cells [9]. Recently, it has been reported that PD-L1 on EVs is highly potent at suppressing immune functions and decreasing IL-2, IFN-γ, and granzyme B production [10]. Interestingly, PD-L1-negative tumor cells and several immune cell types acquire PD-L1 expression from PD-L1-transporting EVs [11]. High levels of EV-bound PD-L1 have been reported as a biomarker for worse prognoses in several cancers [10]. These facts indicate that the removal of sPD-L1 and EVs with PD-L1 may enhance antitumor immune function.

EVs contain nucleotides and proteins, suggesting that they may be involved in cell-to-cell communication [9,12]. Recently, it has been reported that tumor-derived EVs (T-EVs) play an essential role in tumor growth, metastasis, and immune regulation [9,12], in addition to a report of T-EV involvement in drug resistance [13]. T-EVs contain immune-suppressive cytokines, such as TGF-β1 [14], which induce drug resistance [15], in addition to the fact that ADC-resistant cells secrete more EVs expressing the ADC target protein compared to sensitive cells [14]. Additionally, reports have demonstrated higher amounts of ADC on EVs from resistant cells versus sensitive cells [14], indicating that T-EVs can trap ADC on their surface.

Thus, strategies to eliminate T-EVs are considered a promising way to reduce the resistant potential of tumor cells [16]. So far, several reagents have been reported to suppress the generation, trafficking, and uptake of EVs to effect suppression of tumor growth [17]. However, since the safety of these reagents for human application has not yet been definitively demonstrated, they are only currently used in animal models. Although it has been reported that the clearance of EVs was induced by anti-CD9 or anti-CD63 antibodies via macrophages [18], it is still unclear whether anti-CD9 and anti-CD63 antibodies affect normal cells that also express CD9 and CD63 [19,20,21]. 

To combat these limitations, the removal of EVs is an ideal alternative strategy. Hemodialysis can directly remove several compounds from the blood. However, it has been reported that the pore size of dialysis filters is smaller than the EV diameter [22], and the literature indicates that circulating EVs increase after hemodialysis [23]. Plasmapheresis devices (e.g., Hemopurifier^®^) are useful options in this regard as they have been reported to efficiently decrease hepatitis C viral load [24]. Although the size and structural similarities between viral and cancer exosomes indicates the possibility of exosome removal by Hemopurifier^®^ [16], there is a need to accumulate evidence to validate methods for removal of target exosomes. On the other hand, plasma exchange (PE) is a suitable method for eliminating EVs and small proteins, such as sPD-L1, as it has been reported to reduce their numbers [25]. But, in spite of this ostensibly beneficial effect, the utility of PE in EV and sPD-L1 removal and its subsequent effect on tumor treatment with ADCs is still unclear. Thus, we aimed to reduce EVs with a plasma exchange method and investigate their therapeutic effect on tumor treatment with ADCs in a murine model.

## 2. Materials and Methods

### 2.1. Animal Usage and Care

F344/NJcl-rnu/rnu rats raised under specific pathogen-free conditions were purchased from CLEA Japan (Tokyo, Japan), and rats between 6 and 7 weeks of age were used for the experiments. 

### 2.2. Cell Lines

NCI-N87 cells (American Type Culture Collection [ATCC], Manassas, VA, USA) and PC-9 cells (The European Collection of Authenticated Cell Cultures [ECACC], Salisbury, UK) were cultivated in RPMI-1640 (Sigma Aldrich, St. Louis, MO, USA) supplemented with 10% FBS (System Biosciences, Palo Alto, CA, USA) and 1% penicillin-streptomycin (FUJIFILM Wako, Osaka, Japan), at 37 °C under 5% CO_2_.

### 2.3. Rat Tumor Model

Briefly, we intradermally injected 1 × 10^7^ NCI-N87 cells into the dorsal skin of each extremity. When the tumors reached approximately 10 mm in diameter, we performed plasma exchange and administration of T-DM1. Tumor diameters were measured with a caliper, and the tumor volume was determined according to the following formula: tumor volume (mm^3^) = (length) × (width)^2^ × 0.5. 

### 2.4. In Vitro and In Vivo Circuit Assays

The newly developed plasma membrane for the plasma exchange circuit assay was kindly gifted from Toyobo. For the In Vitro Circuit Assay, circuits were filled with heparin-containing saline (10 U/mL). Bovine blood was filtered and centrifuged at 3000 rpm for 15 min, and 50 mL of the supernatant plasma was collected in a centrifuge tube. Prepared plasma was used to fill the circuit up to the 3-way stopper. Next, 2.5 mL of PC-9 or 400 µL of NCI-N87 culture supernatant for the assessment of sPD-L1 or exosomes was added to 5 mL or 20 mL of bovine blood, respectively. To replace displaced liquid, circulation was continued for 82 min (Figure 1A). For the In Vivo Circuit Assay, the circuit was filled with heparin/saline as before (10 U/mL). After rat plasma, derived from tumor-free rats, was thawed at 37 °C, the cryoprecipitate was removed with tweezers. Prepared plasma was used to fill the circuit up to the three-way stopper. Under anesthesia, using isoflurane inhalation, catheters were placed in the femoral artery and vein. After connecting each catheter to the prepared circuit, plasma exchange was started at the speed of 1.2 mL/min. Body temperature was monitored at 36.5 °C using a rectal probe and an automated temperature monitor. Plasma exchanges were performed approximately every 25 min to achieve a 90% plasma exchange rate. Three mg/kg of T-DM1 was administered through the three-way stopper, which was connected to the venous circuit (Figure 1B).

### 2.5. Measurement of sPD-L1 in In Vitro Circuit Assays

To assess the removal of sPD-L1 in the circuit, 100μL of samples collected before and after plasma exchange was used for an ELISA. PD-L1 ELISA was carried out by a human PD-L1 ELISA kit (Abcam, Cambridge, UK) according to the manufacturer’s instructions. OD values were averaged from the result of 3 wells, and the relative decrease was calculated as the OD value after plasma exchange divided by the OD value before plasma exchange. 

### 2.6. Preparation of Exosomes

Collected samples from circuits were applied to qEV single 35 nm columns (Meiwafosis, Tokyo, Japan) to purify exosomes according to the manufacturer’s instructions. Separated fluid was put into tubes with Amicon Ultra filters (Merck Millipore, Burlington, MA, USA) and centrifuged at 14,000× *g* for 10 min at 4 ℃ before Western blotting. 

### 2.7. Western Blotting

Collected exosomes were resuspended in 2x Tris-Glycine SDS Sample buffer and denatured at 95℃ for 5 min. All samples were then subjected to a 4–20% Tris-Glycine mini gel (Thermo Fisher Scientific, Waltham, MA, USA) and electrophoresed at 225 V for 35 min in Tris-Glycine SDS running buffer. Following electrophoresis, the proteins were then transferred onto a PVDF membrane (Thermo Fisher Scientific) and probed with primary biotin-conjugated antibody for CD9 (Clone 30B, rat IgG, FUJIFILM Wako, Osaka, Japan) and streptavidin with the polymerized form of horseradish peroxidase (Thermo Fisher Scientific). The membranes were incubated with ImmunoStar LD (FUJIFILM), and images were subsequently acquired and analyzed for signal intensity using FUSION SOLO 7S (Vilber Bio-Imaging, Collégien, France).

### 2.8. Statistical Analyses

We applied the linear mixed-effects model, including tumor volume, as the outcome with treatment, polynomial in days post-treatment and their interactions as the fixed effects, and rat ID as the random effect. The degree of the polynomial was chosen based on Akaike’s information criterion, and robust, sandwich-type variances were used in the mixed-effect model analyses. Based on this model, marginal means on days 7, 14, and 21 were compared among treatments. Throughout the analyses, *p* values < 0.05 were considered significant. The statistical tests were 2-sided.

## 3. Results

We first assessed the removal of sPD-L1 by filtration with a novel membrane using an in vitro plasma exchange model. Concomitant with another study, we found that plasma exchange decreased the amount of sPD-L1 in the circuit, reaching a reduction of approximately 90% (Table 1). 

Next, we evaluated whether plasma exchange reduced EVs using an in vitro plasma exchange model by western blot. We found that signal intensities before and after plasma exchange were 1444 and 353, respectively, indicating an approximately 75% reduction of tumor-derived EVs (Figure 2). 

Next, this membrane was applied to a murine in vivo plasma exchange model. To evaluate the effect of EV removal during antitumor treatment, we analyzed tumor growth after T-DM1 administration and found that both T-DM1 and combination therapy of T-DM1 plus plasma exchange inhibited tumor growth by day 21, whereas plasma exchange alone failed to suppress tumor growth. There were no significant differences between the T-DM1 and combination therapy groups at the three time points. In addition, the combination therapy significantly reduced tumor size on day 7 compared to the plasma exchange group. Although the T-DM1 group showed tumor reduction on day 7, it was not statistically significant when compared to the plasma exchange monotherapy group (Figure 3) 

## 4. Discussion

We developed a membrane that efficiently removed sPD-L1 and captured EVs. In addition, our data suggest that plasma exchange may potentially enhance the antitumor effect of T-DM1. 

The effect of dialysis on the removal of EVs is controversial as de Laval et al. reported an increase in amounts of large EVs after hemodialysis [23], and Daniel et al. also reported elevated levels of large, neutrophil-generated EVs after hemodialysis [26]. In contrast, Ruzicka et al. showed that submicron particles, ranging from 20–1000 nm, decreased after dialysis [22]. These facts indicate that differences may be due to discrepancies between reported methods, including the type of patients, repetitive mechanical stress from the hemodialysis treatment, and the type of dialyzer. On the other hand, plasma exchange is ideal for removing small compounds since the membrane filters plasma-restricted proteins. Orme et al. reported the efficacy of plasma exchange to remove sPD-L1, EVs, and PD-L1 on EVs [25], showing 70.8%, 33.5%, and 73.1% reductions, respectively. Additionally, multiple plasma exchange procedures are possible for the complete removal of these compounds, suggesting that plasma exchange is feasible for the significant reduction of bloodborne sPD-L1 and EVs. However, the removal of EVs by plasma exchange alone was insufficient to suppress tumor growth in our model. We speculate that this may be due to the single plasma exchange performed in our study. Orme et al. reported that sPD-L1 recovered 33.8% between exchanges, with each plasma exchange typically performed at 1- to 3-day intervals [25], indicating that the concentration of sPD-L1 can theoretically rebound 10 days after plasma exchange. In addition, some patients experienced increased sPD-L1, EVs, or PD-L1 on EVs after the first plasma exchange [25]. Given the amount of EV and sPD-L1 increase in patients with malignancies [25,27], a single plasma exchange may be insufficient. Consistent with our result, Nishida-Aoki et al. demonstrated that the depletion of EVs with anti-CD9 or CD63 antibodies failed to inhibit tumor growth at the primary site despite the suppression of metastasis [18]. In that report, anti-CD9 and CD63 antibodies did not affect vascular generation in the primary tumor, and, moreover, these antibodies did not affect tumor cell proliferation and invasion [18]. These results indicate that EVs might play a more important role in metastasis rather than tumor growth. 

It has been reported that T-EVs express HER2 [28]. Given that soluble HER2 neutralizes anti-HER2 antibodies [29], HER2-positive T-EVs may inhibit the effect of anti-HER2 antibodies, resulting in the enhancement of tumor growth. HER2-positive T-EVs bind efficiently to anti-HER2 antibodies, reducing their amounts on the tumor surface [28]. Because binding of the HER2 motif to T-DM1 is competitive with T-EV-bound HER2, depletion of T-EVs may increase the binding opportunities of T-DM1 to tumor cells, resulting in the enhancement of ADCC by immune cells and inhibition of HER2-mediated growth signal transduction. Collectively, we showed that the tumor volume on day 21 was not different between T-DM1 single treatment and combination treatment; however, compared to T-DM1 alone, the combination therapy seemed to induce the early tumor shrinkage suggestive of positive prognoses [30]. Therefore, our data indicated that plasma exchange with a novel membrane might reduce HER2 positive T-EVs, enhancing the effect of T-DM1.

Since sPD-L1 is also released from non-tumor cells to maintain the immune response, our model system may remove sPD-L1, which is involved in physiological homeostasis of the immune response, leading to dysregulation of autoimmune responses as immune-related adverse events caused by checkpoint inhibitors [31].

Given that tumors continually generate T-EVs and sPD-L1, the concentration of T-EVs and PD-L1 may recover soon after plasma exchange, indicating that multiple plasma exchanges are required to increase the effectiveness of exosome removal. Our model is based on a single plasma exchange within 21 days due to the difficulty of ensuring a catheter insertion. Thus, clinical applications of this system could be challenging. However, the current study has established the utility of plasma exchange enough to serve as a platform for further development. Therefore, hemodialysis patients with tumors may be good candidates for future clinical trials.

Plasma exchange seemed to enhance tumor shrinkage by T-DM1, indicating that our membrane efficiently captured T-EVs in addition to the reduction of several molecules, including sPD-L1, in vivo. Due to the study’s exploratory nature and small sample size, no significant differences were detected between the T-DM1 group and the combination treatment group. Although combination therapy exerted a putative antitumor effect, more data, including the results of multiple plasma exchanges and extension of observation of tumor growth, should be accumulated in the future to test the hypothesis.

## 5. Conclusions

In conclusion, we have proposed a novel therapeutic strategy to inhibit tumor growth by inhibiting the treatment-resistant function of T-EVs with a novel plasma exchange membrane. This strategy can be applied to enhance various treatments against diverse T-EV-generating cancer types.

## Figures and Tables

**Figure 1 biomedicines-10-02483-f001:**
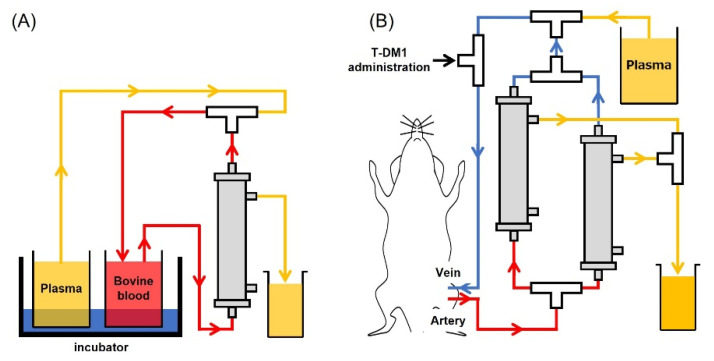
Schematic figure of plasma exchange. (**A**) In vitro plasma exchange circuits, fluid from the bovine blood pool recovers through the plasma membrane unit. (**B**) In vivo plasma exchange circuits, femoral vein, and artery were connected to the catheter. Blood was drawn (up to 2 units of plasma exchange) and returned to the femoral vein.

**Figure 2 biomedicines-10-02483-f002:**
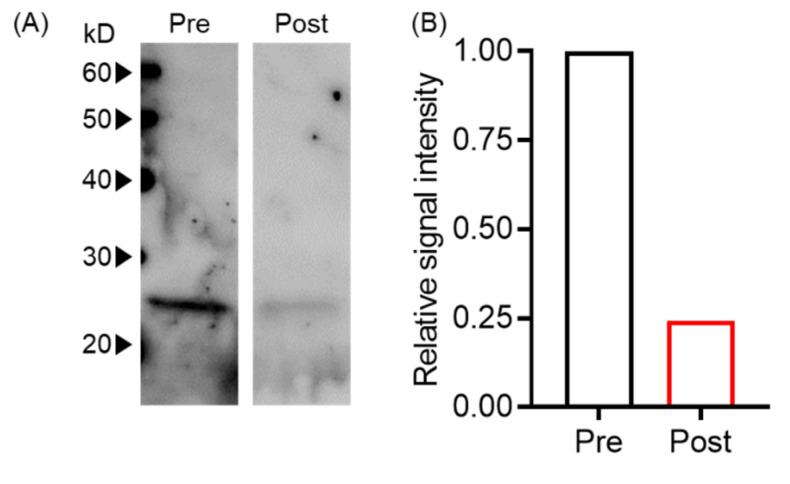
Plasma exchange-reduced EVs. (**A**) T-EV levels pre- and post-plasma exchange were analyzed by Western blot with anti-human CD9 antibody. (**B**) The signal intensity of Western blot bands in (**A**) was normalized with pre-plasma exchange.

**Figure 3 biomedicines-10-02483-f003:**
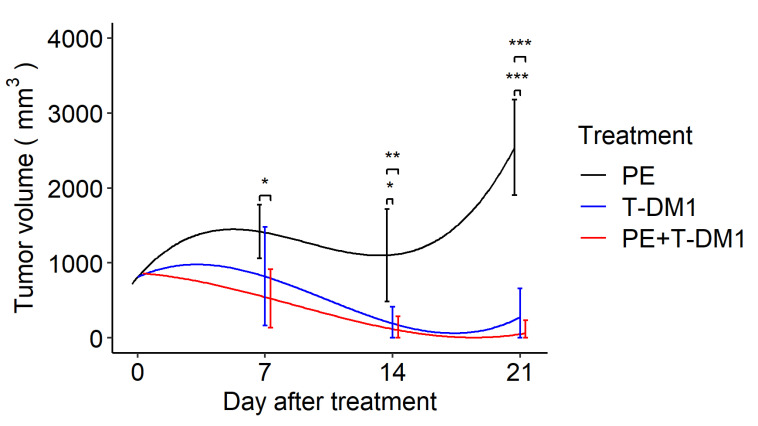
A drug/filtration combination increases the removal of EVs and enhances tumor shrinkage by T-DM1. Longitudinal marginal means of the tumor volume after the tumor size reached approximately 10 mm in diameter with treatment by plasma exchange, T-DM1, and plasma exchange with T-DM1 (*n* = 12 in each group) based on the linear mixed model. PE: plasma exchange, T-DM1: trastuzumab emtansine. Error bars indicate confidence intervals; * *p* < 0.05, ** *p* < 0.01, *** *p* < 0.001.

**Table 1 biomedicines-10-02483-t001:** The OD value of sPD-L1 in the in vitro circuit before and after plasma exchange.

	OD Value			Average
Before	0.12	0.13	0.12	0.123
After	0.01	0.02	0.01	0.013

## Data Availability

The datasets used and/or analyzed during the current study are available from the corresponding author on reasonable request.

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
