# Peer review of "Plasma Exchange May Enhance Antitumor Effects by Removal of Soluble Programmed Death-Ligand 1 and Extracellular Vesicles: Preliminary Study"

_biomedicines, 2022, doi:10.3390/biomedicines10102483_

Round 1

Reviewer 1 Report (Previous Reviewer 1)

The authors did an excellent job addressing comments in the manuscript.  The one remaining issue is that the approach does not appear to be practical for clinical use given our current understanding of normal and pathogenic extracellular vesicle functions and heterogeneity and the existing technology available in the clinic at this time to successfully produce results.

Author Response

We really appreciate the helpful comments and suggestions by the referees who acknowledged our paper as interesting. The following (in red script) are our responses to the comments by the reviewers.

Reviewer 1

The authors did an excellent job addressing comments in the manuscript.  The one remaining issue is that the approach does not appear to be practical for clinical use given our current understanding of normal and pathogenic extracellular vesicle functions and heterogeneity and the existing technology available

 in the clinic at this time to successfully produce results

We are thankful for this important comment from Reviewer 1. As you pointed out, the concept of removing soluble PD-L1 and exosomes by plasma exchanges could be challenging in clinical settings so far. However, the number of studies about exosomes has increased, which could unveil the nature of exosomes in normal and pathogenic functions in the future. In addition, the progress of medicine could be achieved by improving devices by the need to combat the clinical challenges. Collectively, it would be worth proceeding with our studies, even though it seems non-practical now. By contrast, it should be necessary to note the limitation. Therefore, we have added the limitation in the discussion. (Page 7, line 242-244)

Reviewer 2 Report (Previous Reviewer 2)

The authors addressed all the minor issues and Question 1,2,3 I raised previously, and the manuscript looks better now.

However, there are still several things that needs to be addressed before considering for publication:

1. For Quantification of band intensity of WB, the author can use a bar graph and showing the fold decrease post-plasma exchange normalized to pre-plasma exchange instead of directly mention the exact band intensity numbers.

And I still have some concerns about Fig.3 which makes the key conclusion of the manuscript obscure.

1.The purpose of the study is to understand whether PE removal of PD-L1 can inhibit tumor growth. However, in the current figure, there is no basal level of tumor growth (Pre-PE) to compare to (Pre-PE vs PE). So right now, it is not clear if PE itself can help with tumor inhibition.

2.There’s no statistics and p value comparing between the TDM-1 vs PE+TDM-1 group at Day 7, 14 and 21, and it looks to me that even at Day 7 there is no significant difference. These data indicated that PE has no additional effect upon TDM-1 treatment.

Therefore, based on the current figure 3, The statement that “plasma exchange with this novel membrane could achieve early tumor shrinkage” is still really shaky. The only significant data from the figure is that TDM1 treatment can inhibit tumor growth compared to PE group, but these two conditions are not comparable with each other directly.

In my opinion, the authors need to add Pre-PE group in the figure and comparing the statistics between four groups instead of three (Pre- PE, PE, TDM1, PE+TDM1) to make this part of the manuscript convincing.  

Author Response

We really appreciate the helpful comments and suggestions by the referees who acknowledged our paper as interesting. The following (in red script) are our responses to the comments by the reviewers.

Reviewer 2

The authors addressed all the minor issues and Question 1,2,3 I raised previously, and the manuscript looks better now.

However, there are still several things that needs to be addressed before considering for publication:

  1. For Quantification of band intensity of WB, the author can use a bar graph and showing the fold decrease post-plasma exchange normalized to pre-plasma exchange instead of directly mention the exact band intensity numbers.

We are thankful for this Reviewer 2’s comment. We have modified the figure to be normalized on pre-plasma exchange. (Page 5, line 172-174)

And I still have some concerns about Fig.3 which makes the key conclusion of the manuscript obscure.

1.The purpose of the study is to understand whether PE removal of PD-L1 can inhibit tumor growth. However, in the current figure, there is no basal level of tumor growth (Pre-PE) to compare to (Pre-PE vs PE). So right now, it is not clear if PE itself can help with tumor inhibition.

We are thankful for this Reviewer 2’s comment. We agree that it is unclear whether PE itself has antitumor effect as you pointed out. It is important to assess whether PE exerts tumor reduction with the experiments based on comparisons in two group between PE and Pre-PE control groups. However, we would like to focus on the efficacy of the removal of PD-L1 and exosomes by PE on ADCs anticancer effect instead of the evaluation of antitumor effect of PE itself. Therefore, we have modified our misleading statement and added the description (Page 2, line 83-86).

2.There’s no statistics and p value comparing between the TDM-1 vs PE+TDM-1 group at Day 7, 14 and 21, and it looks to me that even at Day 7 there is no significant difference. These data indicated that PE has no additional effect upon TDM-1 treatment.

Therefore, based on the current figure 3, The statement that “plasma exchange with this novel membrane could achieve early tumor shrinkage” is still really shaky. The only significant data from the figure is that TDM1 treatment can inhibit tumor growth compared to PE group, but these two conditions are not comparable with each other directly.

We are thankful for this Reviewer 2’s comment. Although there is a tendency that PE+TDM-1 has a superior antitumor effect to the TDM-1 group, unfortunately, it is not statistically significant in comparison to the tumor volume reduction between TDM-1 and PE+TDM-1 group at day 7. Given that not TDM-1 but PE+TDM-1 has a statistically significant in reduction of tumors compared to PE, this could be explained by the fact that the study volume is limited as we discussed. (Page 7 ,line 248-250) Therefore, we have changed the expression “we demonstrated that plasma exchange with this novel membrane could achieve early tumor shrinkage” into “our date suggested that plasma exchange may potentially be capable of enhancing antitumor effect of TDM-1.”. (Page 6, line 193-194)

In my opinion, the authors need to add Pre-PE group in the figure and comparing the statistics between four groups instead of three (Pre- PE, PE, TDM1, PE+TDM1) to make this part of the manuscript convincing. 

We are thankful for this Reviewer 2's opinion. As you indicated, it is important to conduct experiments with four groups to determine the effect of PE on tumor growth. As described above, our study was designed to focus specifically on the combination effect of PE with TDM-1, and we would like to keep the focus intact. We recognized your point as a consideration for future study. Thank you for the suggestion.

Round 2

Reviewer 2 Report (Previous Reviewer 2)

The authors addressed the questions I raised. 

Just for Fig.3, in order to support the revised statement and make the point clearly to be seen, I suggest to add a "n.s" between the PE and TDM-1 group at Day 7. 

This manuscript is a resubmission of an earlier submission. The following is a list of the peer review reports and author responses from that submission.

Round 1

Reviewer 1 Report

The authors present a well-written study examining plasma exchange as a means to remove PD-L1 expressing exosomes.  They use a mouse model for plasma exchange.  However, the rationale for using plasma exchange for the goal of removing PD-L1 expressing exosomes could result in unforeseen detrimental physiological consequences and will not be practical or cost effective for humans for the following reasons. 

1. Plasma exchange is typically used to remove excess autoimmune antibodies that exhibit direct cause and effect relationships via initiating inflammatory processes.  Instead the authors hypothesized that removing PD-L1 expressing EVs will be therapeutic for cancer.  The problem is that PD-L1 is a general marker not specific to cancer but also involved in other physiological processes including cardiovascular regulation and balancing a normal immune response to disease.  As a result, plasma exchange will remove all PD-L1 expressing exosomes including non-tumor derived exosomes involved in vascular and immune homeostasis.  This limitation is not addressed in the discussion.

2. As an alternative to plasma exchange, exosomes have been filtered out of cancer patient plasma using a Hemopurifier filter cartridge produce by Aethlon Medical Inc.  This technology has been available for at least a decade.  The Hemopurifier filtration system is in clinical trials with the immune checkpoint inhibitor pembrolizumab.  The Hemopurifier technology is superior to plasma exchange because it does not introduce foreign plasma to immune compromised cancer patients that could further exacerbate alloantibody responses making it difficult to match and obtain alloantibody compatible life-saving blood products in the future.  This alternative technology is not discussed in the manuscript and should have been.

3. For the proposed plasma exchange system to work, multiple rounds of plasma exchange will be necessary.  This increases the risk of cancer patients experiencing dangerous transfusion reactions and is not cost effective or practical.  Further exosomes rapidly bind to cells in the local microenvironment, so it will be difficult to establish a universal protocol for all patients given individual differences in tumor anatomy, exosome production and timing of venous drainage.  This needs to be addressed as a limitation well.

4. The manuscript contains a lot of data that is important.  However, overall the data show negative results in that a single round of plasma exchange is not effective.  However, these results should be reported, and limitations expanded on as described above.  Negative data can be just as important or more important in some cases than positive data since it helps with the reformulation of hypotheses and better focuses new experimental directions.

Reviewer 2 Report

This paper looked into the effect of clearance of serum PD-L1 and extracellular vesicles after plasma exchange. and whether this can have a stronger effect on inhibiting tumor cell growth.  However, the data provided by the authors didn’t seem to support their hypothesis.

Major issues:

1.       Line 35-38: This description of the PD-L1 and PD1 is obscure. Please specify and give a more clear description of PD-L1 and the relation of it between PD-1. Like where are they? how do they respond to each other?  Does the “binding of PD-L1” mean using anti-PDL1 to block the function of PD-L1 on tumor cells? please make it precise.

2.       Line 63: Did the reference talk about the increased ADC number? This citation here is only mentioning about increased immune response. Please double check if this is the correct citation here.

3.       Table 1: What is the average OD value after plasma Exchange? The average number didn’t seem to be correct.

4.        Fig 2A and 2B: Please provide a cleaner WB membrane; the background has lots of non-specific background; For quantification, either provide statistics with a combination of at least 3 independent experiments for bar graph or just provide a relative band intensity for this batch of experiment.

5.       Fig.3: please provide a representative of picture for a series of tumors from these groups.

6.       Line 189-191: The data provided for this figure don’t support this statement. Based on the data in Fig.3, the group after Plasma exchange didn’t strengthen the effect compared to only T-DM1 treated group and there’s no significant difference.

Minor issues:

1.       Line 144: is this PVDF membrane?
